# Non-Alcoholic Fatty Liver Disease or Type 2 Diabetes Mellitus—The Chicken or the Egg Dilemma

**DOI:** 10.3390/biomedicines11041097

**Published:** 2023-04-04

**Authors:** Marcin Kosmalski, Agnieszka Śliwińska, Józef Drzewoski

**Affiliations:** 1Department of Clinical Pharmacology, Medical University of Lodz, 90-153 Lodz, Poland; 2Department of Nucleic Acids Biochemistry, Medical University of Lodz, 92-213 Lodz, Poland; 3Central Teaching Hospital of Medical University of Lodz, 92-213 Lodz, Poland

**Keywords:** non-alcoholic fatty liver disease, type 2 diabetes mellitus, coexistence, epidemiology, diagnosis, pathogenesis

## Abstract

In clinical practice, we often deal with patients who suffer from non-alcoholic fatty liver disease (NAFLD) concurrent with type 2 diabetes mellitus (T2DM). The etiopathogenesis of NAFLD is mainly connected with insulin resistance (IR) and obesity. Similarly, the latter patients are in the process of developing T2DM. However, the mechanisms of NAFLD and T2DM coexistence have not been fully elucidated. Considering that both diseases and their complications are of epidemic proportions and significantly affect the length and quality of life, we aimed to answer which of these diseases appears first and thereby highlight the need for their diagnosis and treatment. To address this question, we present and discuss the epidemiological data, diagnoses, complications and pathomechanisms of these two coexisting metabolic diseases. This question is difficult to answer due to the lack of a uniform procedure for NAFLD diagnosis and the asymptomatic nature of both diseases, especially at their beginning stages. To conclude, most researchers suggest that NAFLD appears as the first disease and starts the sequence of circumstances leading ultimately to the development of T2DM. However, there are also data suggesting that T2DM develops before NAFLD. Despite the fact that we cannot definitively answer this question, it is very important to bring the attention of clinicians and researchers to the coexistence of NAFLD and T2DM in order to prevent their consequences.

## 1. Introduction

Among the types of chronic liver disease, non-alcoholic fatty liver disease (NAFLD) is the most frequent. At the same time, NAFLD seems to be underestimated because it has a divergent clinical picture and there is no unified definition and procedure to diagnose the disease, which finally affects therapeutic options. The definition of NAFLD varies with the ways of diagnosing this disease. These include methods for diagnosing fatty liver infiltration (laboratory tests, imaging, biopsy, risk factors), the exclusion of secondary causes of NAFLD and the extent of the affected liver parenchyma [1]. According to the current definition, NAFLD is recognized when more than 5% of hepatocytes are overloaded by lipids based on the imaging or histopathological examination [2]. NAFLD is a biologically and clinically heterogeneous condition. Most patients suffering from NAFLD show isolated steatosis (non-alcoholic fatty liver, NAFL). A smaller percentage of patients have signs of steatohepatitis (NASH) that frequently leads to progressive fibrosis, cirrhosis, hepatocellular carcinoma (HCC) and ultimately to death. Many clinicians treat NAFLD as one of the components of metabolic syndrome (MS), with particular emphasis on the coexistence of this pathology with obesity and impaired glucose and/or lipid metabolism [3,4]. This approach led to the introduction of the concept of metabolic-associated fatty liver disease (MAFLD) [5,6]. 

Among diabetologists, there is current discussion on the definition of MAFLD since metabolic disorders are the main cause of NAFLD. Several experts have defined MAFLD as the relationship of fatty liver disease with obesity, type 2 diabetes mellitus (T2DM) or at least two factors connected with the symptoms of metabolic dysfunction including elevated waist circumference (WC), increased serum C-reactive protein, prediabetes (pre-DM), increased blood pressure, reduced level of high density lipoprotein (HDL-CH), and increased level of triglycerides (TG). They also emphasize that the diagnosis of MAFLD no longer requires the exclusion of alternative causes of chronic liver disease such as alcohol or viral hepatitis [7]. It is noteworthy that the American Society for the Study of Liver Diseases (AASLD) and the European Association for Liver Research (EASL) commented on the proposed MAFLD definition. Firstly, some concerns were expressed about unclear selection of panel experts and the process of formulation of the new definition. Secondly, the methodology of selection of studies considered in the formation of new definition was not precisely specified. Finally, it was recommended that academic and clinical community should participate in the process of the new definition formation [8]. Although it is suggested that experimental and clinical evidence are strong, the MAFLD definition is not widely accepted [9]. Controversies related to the new definition of NAFLD/MAFLD result from consideration of all causes leading to excessive fatty infiltration of the liver (for example, susceptibility due to gene polymorphism changes, obstructive sleep apnea syndrome, polycystic ovary syndrome). In addition, discussion also arises from the need to take into account alcoholic fatty liver disease (ALD) as a protective factor for NAFLD and the need to maintain alcohol abstinence in NAFLD patients. For this reason, it is suggested that more than 10% of patients with MAFLD are not diagnosed as suffering from NAFLD. It should, therefore, be highlighted that there are still many unknowns in the discussion related to the definition of NAFLD/MAFLD and it requires further research [10].

Although NAFLD/MAFLD is usually associated with obesity, accumulating evidence indicates that this pathology is also present in subjects with normal body weight. This suggests that other factors may exert a significant impact on the development of NAFLD [11]. NAFLD diagnosed in people with a body mass index (BMI) lower than 30 kg/m^2^ is currently called “lean-NAFLD”. However, since body weight is not a component of the NAFLD diagnostic criteria, and the definition of NAFLD as a disease accompanied by lean figure is imprecise, the term “lean-NAFLD” has been suggested as a more accurate description of the condition. It is underlined that NAFLD is uncommon in lean individuals and its pathophysiology remains poorly recognized. Although some features of NAFLD are common in obese and lean NAFLD patients, not all lean NAFLD patients exhibit metabolic factors predisposing them to liver dysfunction. In lean patients, factors such as diet composition, lifestyle and genetic susceptibility are considered as likely to participate in the development of NAFLD. Despite possibly different etiologies of NAFLD in obese and lean subjects, it is thought that the lack of excess body fat does not provide protection from hepatitis, fibrosis, or cirrhosis. Some research has shown an even more severe histological picture and raised mortality from NAFLD among patients presenting normal BMI compared with patients with increased BMI. The fact that NAFLD has no evident symptoms at an early stage combined with normal laboratory and anthropometric measurements possibly blinds clinicians to search for NAFLD in lean subjects [12]. It should be highlighted that lean patients suffering from NAFLD are at higher risk of NAFLD progression to severe liver disease resulting in increased risk of death, which additionally stresses the importance of promoting the awareness of NAFLD in lean individuals [13]. 

T2DM, characterized by chronic hyperglycemia, evolves from insulin resistance (IR) and beta cell dysfunction. In its natural course, insulin secretion increases temporarily accompanied by secondary IR and progressive loss of beta cell mass [14]. Metabolic disturbances accompanying obesity and dysfunction of the musculo–hepatopancreatic axis finally lead to the development of T2DM [15]. Although T2DM is strongly associated with IR and obesity, three phenotypes of T2DM have been described based on body weight. They are obese (BMI > 25 kg/m^2^), patients with normal weight (BMI 18–25 kg/m^2^) and lean T2DM (BMI < 18 kg/m^2^). The last two groups include non-obese individuals, and their thinness is not caused by the disease itself or any other pathological factors. These three T2DM phenotypes may result from gender-associated physiological processes or different metabolic defects. It should be emphasized that some anthropometric indicators do not reflect these differences. Namely, the BMI value does not take into account the sex determined body fat distribution or the age-related decrease in muscle mass. Thus, BMI scores involve some inaccuracy in obesity assessment in particular groups of patients. A subject with abdominal obesity may have a normal BMI and simultaneously have a high risk of death. Although BMI correlates linearly with the IR degree, it does not account for a certain degree of stoutness in various populations and ethnic groups due to different body proportions (i.e., body weight vs. body shape) [16]. In addition, morbid obesity (BMI greater than the 99th percentile) occurring in childhood and adolescence has been found to increase the risk of T2DM in adolescents and young adults. T2DM, similar to NAFLD, is usually asymptomatic at an early stage and frequently is recognized by an incidental blood glucose level measurement. Blood glucose screening in obese subjects is more effective in identifying overlooked T2DM than screening in the general population. It has been demonstrated that in T2DM, IR appears firstly in the muscles of lean individuals predisposed to DM before they become fat. Therefore, the emerging IR is not secondary but plays a major role in excessive fat accumulation related to T2DM. What is more, this early muscular IR is responsible for the development of hyperlipidemia and excessive fat accumulation associated with T2DM [17]. 

The relationship between liver fat infiltration, IR and T2DM is confirmed by numerous epidemiological data, which confirm the coexistence of NAFLD and T2DM. Thus, it is necessary to diagnose T2DM in each patient with NAFLD and vice versa [18]. Moreover, the need to identify patients suffering from both NAFLD and T2DM is especially important due to the fact that both pathologies exert mutual impact on the progression and appearance of life-threatening complications [19]. The major complications of diabetes include micro and macrovascular complications that significantly deteriorate the quality of life and if left untreated, markedly elevate the risk of death. In turn, in some patients, the progression of NAFLD to NASH, cirrhosis and HCC finally leads to liver transplantation. In turn, HCC is at present the second leading reason for shortened life among all cancers worldwide. Other causes of death of patients suffering from NAFLD vary depending on the medical condition. Patients with cirrhosis exhibit mainly liver-related events such as de novo ascites, spontaneous bacterial peritonitis, esophageal varices, hepatorenal syndrome, hepatic encephalopathy, HCC, liver transplantation, and death [20]. Those without cirrhosis mainly suffer from vascular events and non-HCC cancer. Therefore, it should be emphasized that cardiovascular diseases (CVD) are the leading cause of death in patients suffering from NAFLD [3,4]. In turn, MAFLD, similar to NAFLD, increases the risk of death due to end-stage liver failure and HCC, liver transplantation, and the risk of development of CVD and cancers other than liver cancer. Other common life-threatening complications of NAFLD include extrahepatic tumors, liver-related end-stage complications, chronic kidney disease and T2DM [11]. Thus, it seems that MAFLD and NAFLD have a comparable clinical picture of complications and long-term outcomes. It should be emphasized that the increased liver-related mortality among patients suffering from NAFLD is due to IR, but among MAFLD patients, it is mainly driven by ALD. An analysis of the frequency of fatty liver among participants of the NHANES III and NHANES 2017–2018 studies showed that over a period of up to 27 years of follow-up, no differences in cumulative mortality from any cause or specific causes were observed between the NAFLD and MAFLD groups. However, it was revealed that the degree of fibrosis and IR were predictors of liver mortality in the NAFLD group and ALD in the MAFLD group [21]. 

### Aim of the Review

Obesity and IR play a crucial role in the pathogenesis of NAFLD and T2DM, metabolic diseases that frequently coexist. However, the mechanisms of NAFLD and T2DM coexistence have not been fully elucidated. Considering that both diseases and their complications are widespread and significantly affect the length and quality of life, we aimed to determine which of these diseases appears first, thereby showing the urgent need for their diagnosis and treatment. 

## 2. Epidemiological Data

According to the International Diabetes Federation, it was estimated that the number of patients in the world suffering from diabetes exceeded 536 million among adults aged 20 to 79. T2DM is the most frequent type of diabetes. It accounts for over 90% of all cases [22]. The disease affects both women and men, and its incidence increases with age. The incidence of diabetes is the highest between 55 and 59 years of age, but as the obesity epidemic grows, the incidence of the disease is expected to increase, especially in younger age groups (≤40 years) [23,24]. According to Taylor’s analysis of the United Kingdom Prospective Study, approximately 36% of patients had a BMI lower than 25 kg/m^2^ at the time of diagnosis of T2DM [25]. Thus, over 60% of T2DM patients were overweight or obese [26].

Based on the American Association of Clinical Endocrinology data, the overall prevalence of NAFLD is about 25%. It is worth mentioning that the prevalence of potentially progressive NAFLD or NASH amounts to 12–14% [2]. However, a meta-analysis by Riazi et al. revealed that NAFLD affected 32.4% of the globe’s adult population. Moreover, the prevalence of this disease has increased significantly recently, from 25.5% in 2005 to 37.8% after 2016. They also found, that NAFLD appears more often in men than in women [27]. It should be emphasized that, as in the case of NAFLD, there are ethnic differences in the frequency of carbohydrate metabolism disorders in T2DM patients. The meta-analysis carried out by Tang et al. of the world’s population revealed the presence of lean NAFLD in 13.11% of the global population, and in 14.55% of the Asian population [28]. Ethnicity also affects the frequency of NAFLD. According to the report by Huang et al., the Hispanic population exhibits a higher prevalence of NAFLD (37.0%). In turn, the non-Hispanic Black population shows decreased prevalence of NAFLD (24.7%) in relation to the non-Hispanic White subjects (29.3%) [29]. In contrast, recent findings from the Centers for Disease Control and Prevention showed an increased incidence of T2DM in American Indians and Alaska Natives (14.5%), non-Hispanic Blacks (12.1%) and individuals of Hispanic origin (11.8%) [30].

Moreover, the degree of metabolic dysfunction depended on body weight, with significantly reduced severity of this dysfunction in lean-NAFLD compared with overweight NAFLD subjects. Interestingly, among lean NAFLD subjects, only 19.56% had diabetes, compared with 45.7% of obese NAFLD subjects [28]. 

Considering MAFLD, it has been demonstrated that its incidence exceeds 38% in the global population, of which 5.37% present the lean phenotype and 29.78% the non-obese phenotype. Metabolic complications such as DM have also been closely associated with MAFLD patients, both lean and non-obese [31]. Among patients with MAFLD, it was shown that approximately 20% of them had T2DM, and over 57% had signs of MS. A higher incidence of MAFLD in men than in women has also been observed. Interestingly, the prevalence rates of MAFLD using the classic NAFLD diagnosis and the new diagnosis are comparable [32]. 

Epidemiological data concerning NAFLD and T2DM coexistence are not consistent, with estimates ranging from 30 to even 80% of patients (average 55.5%). This discrepancy results from the tools used to diagnose fatty liver infiltration, difficulties in distinguishing between NAFLD and NASH, the age of the study population, BMI, the duration of diabetes and the degree of the patient’s compliance [33]. The tool used for NAFLD diagnosis significantly affects the percentage of coexisting NAFLD and T2DM. Ajmera et al. employed magnetic resonance imaging (MRI)-proton density fat fraction as a tool for diagnosing liver infiltration and found a 65% coexistence rate of T2DM and NAFLD among population older than 50 years [34]. In turn, using computed tomography (CT) as a diagnostic tool, Zhou et al. found that approximately 58.67% T2DM patients also suffered NAFLD [35], and Yamane et al. reported signs of NAFLD in 25.2% of T2DM patients [36]. Our study revealed ultrasound features of NAFLD in 70% of patients during the diagnosis of diabetes [37]. It is noteworthy that the majority of epidemiological studies assess the coincidence of NAFLD and T2DM with fairly divergent mean BMI values and glycated hemoglobin (HbA1c) percentages [38,39,40]. Referring to these discrepancies, it should be emphasized that both BMI and HbA1c are predictors of NAFLD in patients with T2DM [41]. T2DM is diagnosed at an advanced stage when patients report signs such as polyuria, polydipsia and weight loss. The onset of T2DM is known to be asymptomatic. Although carbohydrate metabolism disturbances occur, they are not recognized by the patients. Research shows that this misdiagnosis affects a very high percentage of patients and is caused primarily by the asymptomatic onset of carbohydrate metabolism disorders, as well as by other factors such as the choice of a diagnostic tool, hypertension, non-cash insurance and >10 years of physician’s experience [42,43].

Referring to the aims of this article, there are numerous epidemiological data showing that NAFLD develops before T2DM. A meta-analysis conducted by Mantovani et al. of more than 500 studies published between January 2000 and July 2017 that include more than a year of follow-up found that patients suffering from NAFLD recognized by ultrasonography are more than twice as likely to develop T2DM than those without NAFLD. They also noted that the more severe the steatosis and fibrosis, the higher the T2DM risk [44]. NAFLD was found to be connected with a 2.2-fold elevated risk of developing T2DM. They also documented that the risk of diabetes increases significantly depending on the severity of liver fibrosis and independently of age, sex and obesity rates [45]. Including ultrasonography as a diagnostic tool, it has been shown that NAFLD can predict the risk of T2DM regardless of age and is also a predictor regardless of BMI. It should also be emphasized that increased levels of alanine aminotransferase (ALT), aspartate aminotransferase (AST) or gamma-glutamyl transferase (GGTP) are considered as important predictors of T2DM risk, regardless of age and BMI [46]. In a recently published prospective cohort study involving 365,339 patients with NAFLD without T2DM at the baseline, over the course of approximately 11 years of follow-up, 8774 of these patients developed T2DM. It is worth noting that sleeping 7–8 h a day, the lack of insomnia, no patient-reported snoring, and no frequent daytime somnolence were independently associated with the onset of T2DM, with a 20%, 18%, 16%, and 31% risk, respectively. Approximately 33.8% and 33.5% of T2DM cases in this cohort were attributable to NAFLD and poor sleep patterns, respectively. The risk of T2DM was highest (relative risk 3.17) in subjects with NAFLD and poor sleep patterns. They also noted no significant modification by the sleep pattern (healthy, moderate, and poor) of the correlation between NAFLD and T2DM [47]. A meta-analysis involving 117,020 NAFLD patients over 5 years of follow-up showed an increased risk of T2DM (a combined relative risk of 1.97 for ALT, 1.58 for AST, 1.86 for GGTP, and 1.86 for ultrasonography, respectively) [48]. It has also been reported that increased fatty liver index (FLI) scores significantly elevate the likelihood of occurrence of pre-DM, T2DM and NAFLD in overweight/obese individuals. It was also demonstrated that an overweight/obese NAFLD group with high FLI scores had an almost two times higher probability of pre-DM and 9–10 times higher probability of T2DM development in comparison with an NAFLD group with normal body weight and low FLI scores. Additionally, no differences were observed between the normal weight, low overweight/obesity and low FLI groups in the development of T2DM [49]. In turn, Lee et al. when analyzing a group of patients with pre-DM, revealed that patients with NAFLD were more prone to develop T2DM (adjusted risk 1.81), but this risk increased gradually with tertiles (first −18.0 to −0.4 cm, second 0.0 to 3.5 cm and third 3.6 to 21.0 cm) of WC changes (RRs 1.64, 1.73 and 2.04, respectively) [50]. It is noteworthy that NAFLD, including severe forms, is a stronger risk factor for T2DM development in premenopausal women than in postmenopausal women. This suggests loss of protection from T2DM occurrence in women with premenopausal NAFLD [51]. 

Although numerous data suggest NAFLD as the chicken, there are also data providing evidence that T2DM develops as first. According to Younossi et al., over 55% of adult patients with T2DM have NAFLD, 37.3% have NASH, and 17% have advanced fibrosis. They also showed that the highest incidence of NAFLD in diabetics occurs in Europe (68%). Interestingly, the geographic region and age were related to the occurrence of NAFLD [33]. An observational, descriptive study conducted by Mertinez-Ortega et al. that employed transition elastography (TE) for NAFLD diagnosis found advanced fibrosis in 20% and severe steatosis in more than 50% of patients suffering from T2DM. It should be emphasized that obese people predominated in the assessed population (over 80%). It is worth emphasizing that multivariate analysis showed that lower total bilirubin levels are an independent factor of the highest degree of steatosis (S3) and BMI (with a 30.82–42.57 kg/m^2^ range) for advanced fibrosis in T2DM individuals [52]. An analysis of data from the U.S. National Health and Nutrition Survey included 6727 T2DM patients and 4982 MAFLD subjects. Among the latter group, 2950 patients were diagnosed with NAFLD, but 2032 were not. It should be highlighted that the new definition increased fatty liver diagnosis by 68.89% [53]. Moreover, correlations between NAFLD and T2DM remained strong after adjustment for genetically predicted BMI. Genome-wide association studies indicate that genetic predisposition to T2DM mediated 51.4% of BMI’s effect on NAFLD risk [54]. In a study using the liver fat index (LFAT) in quantitative ultrasound, MS and T2MD, fasting serum insulin, BMI and the AST/ALT ratio were found to be independent predictors of NAFLD. Interestingly, it was noted that for any values of BMI and WC, the Chinese population exhibited significantly increased LFAT than the Finnish population [55]. When analyzing the relationship between NAFLD and varying degrees of carbohydrate metabolism disorders, the results of the ultrasound-based study conducted by Rajpu et al. should be taken into consideration. They observed that the incidence of NAFLD was higher in pre-DM subjects (59%) compared with normoglycemic people (control group) (26%). Hepatic steatosis (first degree of severity) occurred in 37% of patients with pre-DM compared with 22% of controls, and stage 2 occurred in 22% of pre-DM patients compared with 4% of controls. It was also demonstrated that elevated WC and GGTP were the best predictors of NAFLD among people with pre-DM [56]. Research conducted by Chen and Jiang showed that not only higher BMI and IR, but also elevated fasting plasma glucose concentration and TG are the risk factors for T2DM with NAFLD [57]. 

In the discussion regarding the priority of NAFLD and T2DM, the results of a meta-analysis performed by conducted by Hashimoto et al. should be considered. It showed that, compared with metabolically healthy individuals (MHOs) without overweight and fatty liver, the relative risks of incidence of T2DM in the MHO with and without fatty were 3.28 and 1.42, respectively [58].

In contrast to the results presented so far, the research by Labenz et al. rated the prevalence of NAFLD in T2DM patients to be low using the Disease Analyzer Database. The analyzed groups of patients suffering from T2DM with NAFLD and without NAFLD were matched in terms of age, sex, index year, concomitant metabolic diseases, the risk of myocardial infarction, stroke, peripheral arterial disease, chronic kidney disease, and the type of T2DM treatment. They demonstrated that the incidence of NAFLD in T2DM patients diagnosed by primary care in Germany was 7.8%. It is also worth mentioning that patients with NAFLD more frequently received gliptins (+/− metformin) and less frequently insulin within the first year after T2DM diagnosis. They did not observe differences in metabolic compensation (HbA1c range 6.5–7.5%) during follow-up between NAFLD and non-NAFLD patients [59].

Although epidemiological studies mostly suggest the presence of NAFLD as a disease which develops before T2DM, there are also a few studies suggesting the priority of T2DM. In this aspect, the variety of NAFLD diagnostic tools, in contrast to the simple T2DM methods, should be noted. 

## 3. Diagnosis

The emerging need for both diagnosis and treatment of coexisting NAFLD and T2DM is highlighted by the recommendations of world’s diabetes and hepatological societies. In 2016, the European Association for the Study of Diabetes and the European Association for the Study of Obesity pointed out a strong need for NAFLD screening in patients diagnosed with T2DM and vice versa [60]. The current AASLD guidelines indicate the persons with obesity and/or features of MS, as well as patients with pred-DM or T2DM, and those with hepatic steatosis detected by any imaging study and/or persistently elevated plasma aminotransferase levels (for over 6 months) are considered as “high risk” subjects for development of NAFLD and advanced fibrosis and should undergo relevant screening [2]. In turn, the guidelines for the diagnosis of NAFLD in lean people suggest that this pathology should be diagnosed in non-Asian race individuals with BMI < 25 kg/m^2^ and in Asian race subjects with BMI < 23 kg/m^2^. Long et al. indicate also that NAFLD should be considered for diagnosis in lean individuals with metabolic disease (such as T2DM, dyslipidemia and hypertension), elevated liver parameters, or incidentally detected hepatic steatosis. Routine diagnosis of lean individuals with NAFLD for comorbid conditions, such as T2DM, dyslipidemia, and hypertension, is also recommended. It is not advised for lean individuals in the general population to receive routine screening for NAFLD, with the exception of people suffering from T2DM over the age 40. Due to inadequate evidence, testing for genetic variants in patients with lean NAFLD is not advised [61]. 

According to recommendations, imaging studies should be used to diagnose NAFLD, but there are multiple methods of imaging presenting various degree of sensitivity and specificity toward steatosis detection. Liver biopsy remains a conclusive test for the detection of NAFLD; however, this method is associated with the highest percentage of adverse effects for the patient. Thus, liver biopsy is reluctantly approved by patients and not recommended as a method of first choice to diagnose NAFLD. Clinicians frequently express objections to this diagnostic procedure in everyday medical practice, especially with its associated risk and cost, the lack of unified treatment recommendations, and relatively good prognosis for most NAFLD patients [62]. It is noteworthy that some clinicians recommend a biopsy in the case of patients with T2DM, especially in cases with elevated liver transaminase or a high risk of NASH [63]. 

Before starting the procedure of NAFLD diagnosis, the following secondary causes of liver steatosis and fibrosis should be excluded: high alcohol consumption (more than 30 g/day in males and 20 g/day in females), use of certain medications (including HIV-antiretroviral therapy, amiodarone, tamoxifen, glucocorticoids, tetracyclines, and valproic acid), genetic diseases (abetalipoproteinemia, familial hypobetalipoproteinemia, hemochromatosis, Wilson’s disease, familial mixed hypercholesterolemia, glycogen storage disease, Weber–Christian disease, and lipodystrophy), exposure to environmental factors (pesticides, lead, arsenic and mercury), eating and gastrological disorders (severe surgical weight loss, starvation, celiac disease, short bowel syndrome, and total parenteral nutrition) and other causes (chronic HCV infections, polycystic ovary syndrome, hypothyroidism, and amphetamine use) [64,65]. 

A number of imaging modalities and laboratory tests are used to indirectly assess fatty liver infiltration. Imaging studies include ultrasonography, transient elastography (performed using vibrations or FibroScan), CT, MRI and Xenon-133 liver scans. Each of these methods has its advantages and disadvantages, and a specific sensitivity and specificity in diagnosing NAFLD [64,66]. To date, no single imaging method with a high specificity toward NAFLD has been indicated. Therefore, apart from imaging, the non-invasive scales are used. Similar to the imaging methods, the different scales represent various levels of sensitivity and specificity. However, simultaneous use of an imaging method and a scale provides a good tool for NAFLD diagnosis, is safe for patients and is non-invasive, which is very important. As described in Table 1, NAFLD diagnostic methods differ in both sensitivity and specificity. Therefore, we suggest that it may affect the recognition of NAFLD and T2DM coexistence [67].

In addition, it should be mentioned that some patients with NAFLD show progressive disease. Therefore, during the diagnosis of NAFLD, clinicians should pay attention to NASH and the risk of hepatic fibrosis. The fibrosis-4 index (FIB-4) is the preferred initial non-invasive test to assess the risk of liver fibrosis. Plasma liver ALT levels are normal in many patients suffering from NAFLD. Therefore, ALT levels can be unreliable and should not be used alone for NAFLD diagnosis [2]. As a method of assessment of liver fibrosis, FIB-4 is simple, accurate, and inexpensive. It includes the following components: age, platelet count, AST, and ALT. It, therefore, takes into account neither the presence of excess body weight nor carbohydrate metabolism disorders [80]. Despite that, FIB-4 is believed to be an effective method to detect the risk of advanced fibrosis, even in subjects suffering from T2DM [81]. 

Importantly, in lean patients with suspected NAFLD, liver biopsy should be considered if the causes of liver damage and/or the severity of liver fibrosis are uncertain. Other diagnostic procedures such as FIB-4 and imaging techniques (transient elastography and MRI elastography) can be applied as an alternative to biopsy to identify the stage of fibrosis and to follow up the patient. After the diagnosis, these tests should be repeated at 6-month to 2-year intervals, depending on the severity of fibrosis and the patient’s response to the therapeutic procedure [61]. Recently, considerable attention has been paid to the triglyceride glucose index (TyG) as a marker of the risk of NAFLD. The TyG index has been proven to be significantly connected with NAFLD and has demonstrated a better sensitivity for identification of NAFLD risk in comparison with other lipid and glycemic parameters. A logistic regression analysis conducted by Li et al. showed the TyG index is markedly useful for identification of NAFLD in subjects with T2DM (OR 3.27 for NAFLD). In contrast, stratified analysis demonstrated that elevated TyG was more sensitive in younger patients (<65 years; OR 2.35), women (OR 2.69) and patients with BMI < 25 kg/m^2^ (OR, 2.80) for the diagnosis of NAFLD [82]. 

When considering methods for diagnosing NAFLD, it should be noted that some of them take into account the presence of T2DM or typical risk factors for its development. This may erroneously point to T2DM as a pre-NAFLD disease. Moreover, NAFLD diagnosis also requires the assessment of the progression of this disease. Therefore, in looking for an answer to whether NAFLD or T2DM develops first, it is necessary to evaluate their mutual influence on complications.

## 4. Clinical View

A meta-analysis conducted by Jarvis et al. revealed that T2DM increases by more than two-fold the risk of severe liver disease development (including liver cirrhosis, complications of cirrhosis, or liver-related death) in patients with NAFLD. The T2DM data came from 12 studies in which 22.8 million people were observed for a median of 10 years. During this follow-up, as many as 72,792 cases of liver disease development, fatal and/or nonfatal, were recorded. It should be emphasized that the majority of studies included middle-aged people, with seven studies involving men and women in approximately equal numbers, two studies in which only women participated and three studies with only/mainly men involved. In fourteen studies, obesity (BMI > 30 kg/m^2^) was evaluated as a prognostic factor for NAFLD. An analysis of data on 19.3 million participants with a median follow-up period of 13.8 years and 49,541 identified liver-related events showed that obesity moderately increased in the risk of severe liver disease [83]. In a retrospective cohort study involving patients with NAFLD and T2DM with a 5-year follow-up, there was a linear increase in hepatic events (HCC and cirrhotic complications) related to the duration of diabetes, whereas no difference was noted in annual incidence between the first 10 years since T2DM recognition and subsequent years (0.06% vs. 0.10%). Importantly, the multivariate analysis showed that the strongest risk factors connected with hepatic events were the baseline age ≥ 50 years and liver cirrhosis. Therefore, it seems that age, but not the duration of T2DM, is a predictor of hepatic events in patients with NAFLD and T2DM [84]. On the other hand, it has been proven that one in five adult patients with T2DM has increased liver stiffness, as diagnosed by transient vibration-controlled elastography (VCTE). Multivariate meta-regression analysis demonstrated that higher BMI, older age, higher percentage of men, lower VCTE cut-off, and Asian ethnicity are associated with increased rates of prevalence of this pathology [85]. 

In a meta-analysis performed by Song et al., there was no relationship between NAFLD and diabetic retinopathy in people with T2DM. The analysis of subgroups of patients from China, Korea, and Iran showed a reduced risk of diabetic retinopathy among T2DM patients with NAFLD in comparison with those without NAFLD. However, in subjects with T2DM from Italy and India, that risk was increased. Interestingly, the lack of relationship between NAFLD and diabetic retinopathy has been found in subjects from America [86]. From a practical point of view, the results of the study conducted by Zhang et al. seem to be very important. They demonstrated that in the group of patients with T2DM, non-obese patients with NAFLD did not have a better cardio-metabolic risk profile compared with obese people with fatty liver. It is noteworthy that the relationship between metabolic disorders and NAFLD was stronger in female patients with T2DM without obesity than in those with obesity [87]. 

These presented data are not sufficient to conclude that NAFLD or T2DM develops first. Therefore, it should be investigated whether there is a common mechanism.

## 5. Pathogenesis

The existing data suggest considerable complexity in the etiopathogenesis of NAFLD, which is not fully understood. First of all, the major feature of the disease is direct accumulation of fat in the liver resulting from the imbalance between fatty acids (FAs) (originating from the diet, de novo lipogenesis (DNL), and lipolysis of adipose tissue) supplied to the liver, lipid synthesis and oxidation, as well as TG transported from the liver as very low density lipoproteins (VLDL). In the initial stages of NAFLD, there are elevated rates of both VLDL secretion and β-oxidation. The purpose of these processes is to compensate for the raised influx of FAs to the liver. FAs coming from the diet are absorbed from the small intestine, accumulated in the form of chylomicrons and secreted into the blood. Next, the majority of chylomicron reach the adipose tissue where they are stored. The remaining portion of chylomicrons is taken up by the liver. During the postprandial periods, FAs present in the liver come from chylomicrons and chylomicron residues. In turn, fasting FAs are derived from lipolysis of the adipose tissue. It was demonstrated that in subjects with NAFLD, approximately 15% of hepatic FAs come from the diet, 59% from the circulation, and 26% from DNL. The constitution of FAs in the diet can also affect the increase in fat in the liver. FAs present in the liver undergo the following processes: β-oxidation in the mitochondria to form ATP or ketone bodies, esterification to TG lipid droplets in hepatocytes or release into the serum in the form of VLDL particles. Impaired β-oxidation leads to the elevation of lipid content in the liver and IR [88]. 

The main consequence of IR of the peripheral tissues, including the muscles, adipose tissue and liver, is T2DM. IR is believed to precede the development of T2DM by 10 to 15 years. The development of IR usually causes a compensatory increase in endogenous insulin production. Hyperinsulinemia is associated with weight gain which, in turn, exacerbates IR. This vicious cycle continues until the activity of the beta cells of the pancreas is no longer able to adequately meet the insulin demand caused by IR, leading to hyperglycemia. With a constant mismatch between the demand for insulin and its production, glycemia rises to levels consistent with T2DM. There are various mechanisms of chronic hyperinsulinemia contributing to peripheral IR, such as down expression of insulin receptors and changes in signaling cascades, including the inhibition of insulin receptor kinase activity and tyrosine phosphorylation of insulin receptor substrates-1 and -2 (IRS1/2), enhancement of IRS1/2 proteasome-mediated degradation, phosphatase-mediated dephosphorylation and kinase-mediated serine/threonine phosphorylation [89]. Insulin from pancreatic β-cells reaches the liver through the portal circulation to exert its action and is finally degraded the hepatocytes. The latter process, called hepatic insulin clearance, controls the homeostatic level of insulin. The mechanism of insulin clearance involves receptor-mediated insulin uptake followed by its degradation. After phosphorylation by the insulin receptor tyrosine kinase, the carcinoembryonic antigen-related cell adhesion molecule 1 (CEACAM1) becomes a part of the insulin–insulin receptor complex, and thereby increases the rate of its endocytosis and targeting of degradation pathways [90].

The relationship between NAFLD and IR is bidirectional. One the one hand, IR promotes the progression of NAFLD, but on the other hand, NAFLD triggers the development of IR [91]. Hepatic IR, the main component of systemic IR, is also associated with the decrease in insulin sensitivity of skeletal muscle and adipose tissue. Hepatic IR connected with NAFLD is caused by an increased content of hepatic diacylglycerol (DAG), which activates the protein kinase C epsilon type (PKCε). There is a link between DAG-induced PKCε activation in the liver and hepatic IR associated with human NAFLD. DAG localized near the cell membrane activates PKC, which performs multiple phosphorylations, resulting in inhibition of insulin receptor kinase. Consequently, the phosphorylation of tyrosine IRS-1 and -2, inositol 3-kinase (PI3K) are also disrupted, which finally results in insulin signaling attenuation. The ultimate result is a decrease in glycogen synthesis in the liver because of decreased glycogen synthase activation and increased gluconeogenesis resulting from activation of forkhead box O1 protein (FOXO1) that is associated with excessive glucose release via glucose transporter 2 (GLUT2) [92]. Ceramides are also involved in the development of IR. It was demonstrated in rats with NAFLD that inhibition of ceramide synthesis relieves liver steatosis and fibrosis. Moreover, increased hepatic fat content is associated with ceramide-rich liver lipids in “metabolic NAFLD”, but not in “pathatin-like phospholipase containing (PNPLA3) NAFLD” [93]. In contrast, some NAFLD models demonstrate raised levels of DAG and ceramide in the liver without IR. However, these authors conclude that the main effect of PKCε on glucose homeostasis does not have a direct impact on the liver, and cautious interpretation of the activation of PKCε in this tissue is required [94]. There are also data suggesting that phosphatidic acid, rather than DAG, is associated with impaired insulin action in mouse liver cells [95]. Further investigation of the involvement of PKCε in the regulation of insulin signaling in the liver should be conducted. 

AMPK activation plays a key role in metabolic processes. It upregulates glucose uptake, oxidation of FAs, mitochondrial biogenesis, and autophagy and inhibits FAs, cholesterol, and protein synthesis. It was shown that multiple AMPK activators inhibit lipogenesis, reduce lipid content, and improve insulin sensitivity in the liver [96]. Interestingly, patients suffering from NAFLD have lower serum level of adiponectin, a cytokine produced mainly by adipocytes that regulates FA oxidation and inhibits fat accumulation in the liver. It was reported that hypoadiponectinemia during NAFLD development impairs FA metabolism and promotes chronic inflammation in the liver [97]. In addition, hepatokines, such as fetuin A, fetuin B, retinol 4 binding protein (RBP4), and selenoprotein P, were found to participate in NAFLD and IR development [92,98].

Conditions associated with IR such as hyperglycemia and hyperinsulinemia increase the hepatic pool of fatty acyl-CaA [98]. Namely, hyperglycemia induces hepatic DNL through the carbohydrate response element-binding protein (ChREBP), while hyperinsulinemia induces hepatic DNL through sterol regulatory element-binding transcription factor 1c (SREBP1c) [99]. In addition, fructose, a substrate for lipogenesis and glycogenesis, has been identified as a strong enhancer of DNL [100]. It has been also observed that fructose triggers DNL through its metabolism by the intestinal microbiota to acetate, which then reaches the liver through the portal vein [101]. These dual mechanisms may explain the higher lipogenic potential of fructose than that of glucose, which has been observed previously. In addition, the dysregulated intestinal microbiome produces other short-chain FAs, such as butyrate and propionate, which contribute to liver lipogenesis, inflammatory conditions and fibrosis [102,103,104,105]. 

The details of the molecular mechanisms of IR are still being investigated. Recently, endoplasmic reticulum (ER) stress has been identified as a key factor in IT development. Hyperglycemia and the excess of lipids in hepatocytes are a main source of reactive oxygen species (ROS) overproduction leading to ER stress. This condition leads to activation of the transcription factors sensitive to oxidative stress, such as NF-κB. These, in turn, trigger Kupffer cells to produce pro-inflammatory mediators and promote NAFLD progression [106]. In addition, the excess of lipids in hepatocytes results in elevated demand for protein processing by the ER, which causes the accumulation of misfolded proteins in the ER lumen. In consequence, the excess of misfolded or unfolded proteins is a direct inducer of ER stress. The unfolded protein response (UPR) is triggered to restore ER homeostasis by reducing protein synthesis and enhancing protein folding and clearance. Additionally, the circadian clock machinery involved in the regulation of ER stress-related pathways and control of liver metabolism homeostasis has been implicated in NAFLD progression [107]. ER stress is manifested by inflammatory responses, such as direct defense against microbial pathogens, production of pro-inflammatory cytokines, immunogenic cell death, metabolic homeostasis, and maintenance of immune tolerance. In the course of these processes, the liver is infiltrated by immune cells, which release pro-inflammatory cytokines and immunomodulatory mediators that may worsen liver cell dysfunction, resulting in hepatocyte necrosis, hepatic steatosis and fibrosis. Therefore, the progressive fat accumulation in the liver that is characteristic of NAFLD causes lipotoxicity which interacts with ER stress, resulting in inflammation and damage to the hepatocytes and finally NASH [108]. Thus, restoration of ER homeostasis in the hepatocytes of NAFLD patients engages mechanisms that induce persistent ER stress because of the reduced or impaired ability of the immune response to mitigate the inflammation-related damage. Moreover, hyperglycemia and overloading of hepatocytes with lipids are the main sources of reactive oxygen species (ROS) overproduction that triggers ER stress. This condition leads to the activation of transcription factors sensitive to oxidative stress, such as NF- κB. These, in turn, trigger Kupffer cells to produce pro-inflammatory mediators and further promote NAFLD progression [109].

Overproduction of pro-inflammatory cytokines in NAFLD drives IR via NF-κB, which is known as an integrator of the inflammatory pathway response [110]. It regulates the expression of pro-inflammatory cytokines such as tumor necrosis factor α (TNF-α) and interleukin-6-type (IL-6). TNF-α activates the JNK pathway, whereas IL-6 activates the STAT3 pathway, and both pathways inhibit insulin signaling. Other factors, such as microbial lipopolysaccharide (LPS), FFA, advanced glycation endpoints (AGEs), oxidative and ER stress, and inflammatory cytokines, further contribute, through the activation of the NF-κB kinase (IKKβ) β subunit, to attenuation of insulin signaling via JNK activation [111]. Furthermore, NF-κB induces protein transcription of tyrosine phosphatase 1B (PTP1B) and cytokine signaling inhibitor 3 (SOCS3); both affect the phosphorylation of IRS proteins and promote IR [112,113].

IR in NAFLD also has a genetic origin [114]. There is evidence for some degree of heredity of NAFLD. Analysis of the genome has identified a wide variability in phenotypes and the risk of NAFLD progression. The most frequently reported NAFLD-related genetic variants associated with NAFLD have been identified in the genes of PNPLA3, a member of the transmembrane superfamily 6 2 (TM6SF2), glucokinase regulatory protein (GCKR), as well as membrane-bound protein 7 containing the O-acyltransferase domain (MBOAT7), GCKR, and 17-β hydroxysteroid dehydrogenase 13 (HSD17B13). These genes are strongly involved in regulating of mobilization of TG from lipid droplets (PNPLA3), the secretion of LDLs (TM6SF2), hepatic phosphatidylinositol acyl chain remodeling (MBOAT7), de novo lipogenesis (GCKR), or bioactive lipid and estradiol signaling (HSD17B13) [115]. In a large exome-wide association study of plasma lipids involving over 300,000 participants, the genetic variants located in PNPLA3 at the 148Met allele and in TM6SF2 at the 167Lys allele exhibited strong associations not only with liver steatosis and progression to NASH, cirrhosis, and hepatocellular carcinoma, but were also correlated with T2DM, low blood TG, low LDL cholesterol concentration, and protection against coronary artery disease [116]. The association of the I148M variant in PNPLA3 and the E167K variant in TM6SF2 with increased fat content in the liver, but not with the signs of MS and IR, should be emphasized. These genetic forms of NAFLD are predictive of NASH and cirrhosis, but not of T2DM [117]. 

There is also a growing body of evidence suggesting a link between the microbiota–gut–liver axis and IR in NAFLD [118]. Firstly, intestinal dysbiosis drives IR in the liver by bacterial infiltration and production of multiple bacterial metabolites which lead to systemic inflammation. Bacterial infiltration and the related growth of energy acquisition, the production of bacterial metabolites, and the increased gastrointestinal permeability cause intestinal inflammation or immune disorders that lead to systemic inflammation. Bacterial-derived LPS induces a CD14/tool-like receptor (TLR) response that promotes metabolic endotoxemia and leads to inflammation [119]. Secondly, it was reported that alleviated farnesoid x receptor (FXR) activation caused by intestinal microflora dysbiosis is associated with IR and NAFLD [120]. FXR plays an important role in multiple physiological processes, including cholesterol/bile acid (BA) metabolism, glucose/lipid metabolism, and inflammation. After food ingestion, conjugated BA are secreted into the lumen of the intestine. Next, they are broken down by intestinal bacteria and converted to secondary BA which activates FXR and leads to insulin secretion by the pancreas. In turn, primary BA is involved in glucose metabolism by activating TGR-5 and releasing glucagon-like peptide-1 (GLP-1) [121]. It is suggested that the enhancement of FXR activation may protect from NAFLD [117]. A well-recognized contributor to IR and metabolic dysfunction is ectopic fat. The duodenal adipose tissue represents a new and potentially active metabolic site for ectopic fat deposition [118,122]. 

Although it is widely accepted that IR and beta cell dysfunction contribute to the development of T2DM, there is still a debate concerning the sequence of events leading to T2DM [123]. The conventional paradigm is that IR is a primary defect resulting from compensatory hyperinsulinemia and ultimately leading to beta cell depletion and T2DM. However, there is growing support for the theory that hyperinsulinemia may be the first abnormality in the pathogenesis of T2DM, with the main hyperinsulinemic factors being excessive insulin secretion from beta cells and/or decreased hepatic clearance of insulin [124]. In addition to hormones and organokines, blood glucose dysregulation and long-term hyperglycemia in T2DM are associated with metabolism disturbances and production of harmful metabolites in various organs. As suggested by recent data, these toxic metabolites, i.e., FAs, may have negative effects on interorgan communication in the course of T2DM development [125]. It has also been reported that IR is not always associated with a high BMI, and T2DM may result from impaired insulin secretion rather than from IR. It has been suggested that Β cell death and transdifferentiation are responsible for β cell reduction [126]. In T2DM, neogenesis of the pancreatic islets may occur, which compensates for the reduction of β-cell mass and does not affect their proliferation. According to Lytrivi et al., the following factors affect the decrease in β cell volume: blood glucose level, and low incidence or difficult to determine β cell apoptosis. The islet volume itself is not significantly reduced in T2DM [127]. These suggestions support the theory that amylin deposits in fat islets increase the incidence of β cell apoptosis [128]. There is evidence that lipotoxicity can impair beta cell function, especially in individuals predisposed to T2DM. Suleiman et al. emphasized that the beta cell functional damage can recover, provided the metabolic insult is attenuated [129]. However, the short- and long-term effects of lipid infusion in healthy individuals on insulin secretion have been demonstrated. In one study, 24 h after infusion of 10% triglyceride emulsion, fasting plasma non-esterified fatty acid concentrations and the acute insulin response to glucose had returned to baseline values. The reversibility of beta cell functional alterations induced by in vivo “lipotoxicity” is suggested [130].

Amylin aggregates drive inflammatory conditions induced by pro-inflammatory macrophages, leading to β cell dysfunction [131]. Recently, the multifactorial development theory of T2DM has been supported by the gut microbiota theory. The growth of many bacteria, viruses and pathogenic fungi has been demonstrated in patients with T2DM. The presence of these pathogens associated with improper nutrition is correlated with disorders of many proteins and receptors that have direct and indirect effects on insulin secretion, involving the immune response and their metabolites, although this mechanism has not been fully elucidated [132]. Overexpression of pro-inflammatory adipokines, or a lack of anti-inflammatory adipokines in experiments carried out on rodents are causally associated with the occurrence and development of obesity and T2DM. Pro-inflammatory adipokines increase while anti-inflammatory adipokines decrease in both obese rodents and humans, which is associated with corresponding metabolic indicators of obesity and T2DM [133].

Considerable attention has recently been paid to fetuin-A glycoprotein, the main non-phosphorylated plasma sialoprotein. In adults, this protein is released by the liver, placenta and tongue. Abnormally high levels have been detected in patients with severe liver disease. It has been found that fetuin A binds to and inhibits tyrosine kinase of the insulin receptor and, thereby, contributes to the development of IR and consequently of T2DM. It has also been shown to regulate other receptors of growth factors, such as hepatocyte growth factor (HGF) or EGF-containing tyrosine kinase domains. It was demonstrated that increased amounts of fetuin-A secreted by a fatty liver inhibited glucose-stimulated insulin secretion (GSIS) of human islets [134,135,136]. 

The research reported so far points to IR as a crucial factor linking NAFLD and T2DM, and it is interesting whether IR leads first to NAFLD or T2DM. Smith et al. attempted to answer this question. In their study that gradually decreased insulin sensitivity, insulin secretion gradually increased from the lean group with normal glucose tolerance (NL), to obese NL to obese-NAFLD groups, while total liver insulin extraction was higher in obese-NL and obese-NAFLD individuals than in lean subjects. The presence of insulin in the systemic circulation and extrahepatic insulin extraction gradually increased from the lean NL group to obese NL groups to obese NAFLD groups. Total hepatic insulin extraction remained stable at high insulin delivery rates, and the relationship between systemic insulin appearance and total extrahepatic extraction was linear. This means that the greater increase in plasma insulin concentration in response to oral glucose challenge in overweight and obese individuals with NAFLD compared with those observed in the lean NL group is due to increased insulin secretion rather than to a reduction in total insulin extraction by the liver or extrahepatic tissues. In obese NL and obese NAFLD, hyperinsulinemia after glucose intake is caused by an increase in insulin secretion, without a decrease in overall hepatic or extrahepatic insulin extraction. However, the maximum ability of the liver to remove insulin is limited due to the saturable extraction process. The above implies that it is impossible to compensate for the increase in IR, leading to the impaired glucose homeostasis by increased insulin supply to the liver and extrahepatic tissues in obese individuals with NAFLD [137].

In turn, in the study by Ortiz-Lopez et al., liver fat was assessed by magnetic resonance spectroscopy (MRS), insulin sensitivity of the liver and muscle by euglycemic clamp with 3-[(3)H]glucose) and IR indices at the liver level (HIR(i) = endogenous glucose production × fasting plasma insulin (FPI)) and adipose tissue (Adipo-IR(i) = fasting FFA × FPI). They found that IR occurs in the adipose tissue, liver and muscle tissue of patients with NAFLD. The muscle and liver sensitivity to insulin was impaired to a similar degree in patients with NAFLD, regardless of the presence pre-DM or T2DM status. Only adipose tissue IR deteriorated in T2DM and correlated with muscle and liver IR severity, and steatosis detected by MRS. Taken together, these findings indicate that adipose tissue IR plays a major role in the severity of NAFLD in patients with T2DM [138]. 

The hypothesis assuming the priority of NAFLD over T2DM and the association with IR is supported by studies assessing the presence of NAFLD in patients with T1DM, who frequently need higher doses of insulin in the course of diabetes, mainly due to an unhealthy lifestyle leading to weight gain and aggravation of IR. Patients with T1DM usually present as young, thin individuals. A study conducted by Grzelka-Woźniak et al. revealed NAFLD I in 43% of patients with T1DM. Additionally, they found that patients with concomitant NAFLD were less sensitive to insulin (estimated glucose distribution rate (eGDR)) and had a higher visceral adiposity index (VAI). Moreover, the authors observed that indirect IR markers such as eGDR, VAI and TG/HDL-C ratio, adjusted for sex, diabetes duration and HbA1c, were independently associated with NAFLD [139]. In turn, a study by de Vries et al. involving multivariable logistic regression analysis demonstrated that the estimated glucose disposal rate (eGDR) was independently related with the presence of NAFLD after adjustment for the duration of diabetes [140].

Comparing the effect of the entero–pancreatic axis and the incretin system on the progressive relationship between NAFLD and carbohydrate disorders, it is necessary to mention the results of a study conducted by Junker et al. They demonstrated that the healthy subjects exhibited a higher incretin effect (55%) compared with nondiabetic NAFLD patients (39%), NAFLD patients with T2DM (20%), and patients with T2DM and no liver disease (2%). They also found fasting hyperglucagonemia in NAFLD patients with and without T2DM. Fasting glucagon levels were lower but similar in patients with T2DM and without liver disease and in controls. All the groups had similar glucagon-like peptide-1 and glucose-dependent insulinotropic polypeptide responses [141]. It was further shown that in patients with NAFLD, the hepatic/renal echo intensity ratio (H/R) correlated positively with fasting plasma glucose and insulin concentrations as well as OGTT, HOMA-IR, β cell function, and plasma IL-4, IL-17, IFN-γ, TNF-α, FGF and GCSF concentrations. They also observed a negative link between H/R and insulin action evaluated by the 180 min oral glucose insulin sensitivity method (OGIS). Moreover, the multiple stepwise regression analysis revealed that the best H/R predictors were OGIS and postprandial GLP-1, HDL-CH cholesterol, IFN-γ values. The higher predictive value of postprandial variables suggests that liver fat is essentially a postprandial phenomenon, with GLP-1 probably playing a significant role [142]. 

Taken together, these findings suggest there are relationships between microbiota dysbiosis, fat content and IR. Although not all the mechanisms are fully recognized, it is suggested that changes in microbiota composition can induce obesity. One of the mechanisms is the ability of the intestinal bacteria to remove more energy from the diet, probably due to very efficient dietary nutrient degradation by the enzymes produced by such bacteria. There are also data indicating important links between the microbiota, insulin sensitivity and LPS, SCFA, bile acids and BCAA [143]. The relationship between the gut microbiota and the progression of NAFLD to T2DM is unclear. The available data suggest that NAFLD patients, in comparison with NAFLD and T2DM patients, have a different composition of gut microbiota compared with patients with NAFLD or T2DM alone, and this may be associated with development of the disease. In the Leylabadlo et al. study, it was found that the Bacteroidetes and Firmicutes phyla strains were significantly lower in NAFLD patients with T2DM, whereas the Proteobacteria and Actinobacteria contents were higher in NAFLD patients with T2DM, and there were no significant differences between patients from the NAFLD group and the T2DM group. In addition, the counts of Firmicutes copies were lower in the separate NAFLD and T2DM groups compared with healthy controls [144]. 

Fetuin-A is an important glycoprotein that promotes IR by acting as an antagonist of the insulin receptor tyrosine kinase in the liver and skeletal muscle. High levels of fetuin-A in humans have been associated with a higher risk of T2DM and metabolic syndrome [135,145]. When analyzing the cause-and-effect relationship between NAFLD and T2DM, fetuin-A was shown to interfere with the functional maturity of beta cells. It was found that a perinatal decline in fetuin-A attenuated TGFBR signaling in the islets responsible for the functional maturation of neonatal beta cells. Their functional maturity remains revocable later in life, and metabolically unhealthy conditions such as fatty liver and elevated plasma fetuin-A levels were found to impair both beta cell function and adaptive proliferation [146]. In a study evaluating human pancreatic fat cells crosstalk with the islets and the role of diabetogenic factors, fetuin-A was shown to induce mRNA expression of IL6, CXCL8, and CCL2 in isolated primary pancreatic preadipocytes and differentiated adipocytes. Fetuin-A production was dependent on the toll-like receptor 4 (TLR4) and further potentiated in preadipocytes when cultured with the islets. In the islets, IL6 and CXCL8 mRNA levels were increased by fetuin-A. Only in macrophages in isolated islets did fetuin-A stimulate the production of the cytotoxic cytokine IL-1β. Fetuin-A had no pro-apoptotic effect in islet cells, whereas it impaired glucose-induced insulin secretion in a manner independent of TLR4, but dependent on c-Jun N-terminal kinase- and Ca2+. The above suggests the potential contribution of fetuin-A-mediated metabolic crosstalk of fatty liver with islets to obesity-linked blindness of beta cells to glucose, while fatty pancreas may exacerbate local inflammation [147].

Considering the aim of this study, there are studies indicating that T2DM develops before NAFLD. The results of the study by Seeberg et al. suggest that T2DM develops earlier than NAFLD. Patients with T2DM and extreme obesity had high levels of fat liver fraction. In addition, hepatic steatosis, but not the degree of liver fibrosis, was observed to be connected with various parameters of insulin sensitivity in very obese patients with T2DM. This suggests that the liver fat fraction is primarily associated with hepatic, but not peripheral, insulin sensitivity [148]. 

The relationship between IR and the development of NAFLD in patients with T2DM is supported by the results of studies evaluating risk factors for the presence of fatty infiltration of the liver in this patient group. The evaluation of the usefulness of two surrogate IR markers (triglyceride and glucose index (TyG) and the ratio of TG to high-density lipoprotein cholesterol (TG/HDL-C)) in diagnosing MAFLD showed that FPG is a distinct risk factor for MAFLD after adjustment for age, sex and BMI [149]. It has been proven that in the liver of obese patients with T2DM, the increased PDGF-AA (encoding platelet-derived growth factor α) signaling contributes to IR [150]. Studies evaluating plasma miRNAs in patients with or without T2DM complicated by NAFLD showed that the plasma levels of miR-17, miR-20a, miR-20b, and miR-122 were elevated in T2DM patients with NAFLD compared with patients without NAFLD. Animal studies on rats further demonstrated that these miRNAs were more sensitive than the traditional serological markers in predicting complications [151]. In regard to the suggested involvement of the inflammatory response in the development of NAFLD and T2DM, it was found that an increase in free radical-induced oxidation, TNF-α and NF-κB, and depletion of the antioxidant system appear to be the key factors in the development of NAFLD in patients with T2DM [152]. However, in a study of adult Greeks, it was shown by multivariate analysis that the PNPLA3 variant rs738409, WC and female sex were directly associated with fatty liver disease, whereas the duration of DM had an inverse association [153]. The available data also indicate a key relationship between GABA production and transport in the liver and insulin activity, HOMA-IR, T2DM, and BMI [154]. 

The theory of the priority of T2DM development over NAFLD is also supported by research confirming the presence of a relationship between the intestinal microflora and the severity of NAFLD in patients with T2DM. Tsai et al. found that out of 163 patients with T2DM, 83 patients with moderate to severe NAFLD had higher Firmicutes counts compared with 80 patients without NAFLD or with mild NAFLD. The severity of NAFLD in T2DM patients was increased by high Firmicutes content. A positive correlation between the severity of NAFLD and the Firmicutes type was found in men with T2DM with a body mass index of ≥24 kg/m^2^ and glycated hemoglobin <7.5%. Enrichment of fecal microbiota with the Firmicutes type correlates significantly and positively with the severity of NAFLD in patients with T2DM [155].

The pathophysiological relationship between NAFLD and T2DM is presented in Figure 1.

## 6. Conclusions

There is a strong bidirectional relationship between NAFLD and T2DM that is confirmed by epidemiological data, clinical picture, diagnosis and pathomechanisms. Most researchers and clinicians indicate NAFLD as the metabolic pathology that emerges first and initiates the sequence of events leading to the development of T2DM. The relationship between NAFLD and T2DM undoubtedly involves a complex causal link between these two metabolic diseases, with IR, however, being their common root. The still unresolved controversies regarding the causal relationship between NAFLD and T2DM do not relieve the clinician of the thorough search for these diseases in a patient with an initial diagnosis of one of these entities. The right diagnosis at the right time, rather than scientific discussions, is of great importance to the patient.

## Figures and Tables

**Figure 1 biomedicines-11-01097-f001:**
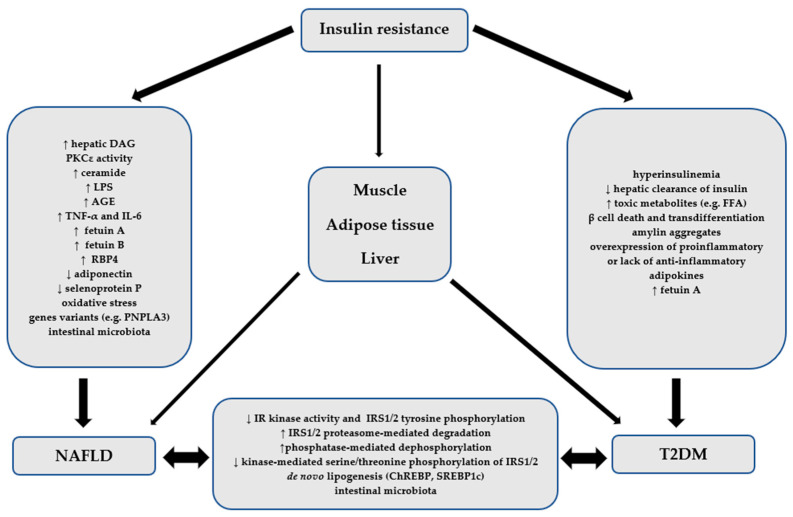
The pathophysiological relationship between NAFLD and T2DM. AGE—advanced glycation endpoint, ChREBP—carbohydrate response element-binding protein, DAG—diacylglycerol, FFA—free fatty acids, IL-6—interleukin-6-type, IR—insulin receptor, IRS1/2—insulin receptor substrates-1 and -2, LPS—lipopolysaccharide, NAFLD—non-alcoholic fatty liver disease, PNPLA3—pathatin-like phospholipase containing 3, PKCε—protein kinase C epsilon type, RBP4—retinol 4 binding protein, SREBP1c—sterol regulatory element-binding transcription factor 1c, T2DM—type 2 diabetes mellitus, TNF-α—tumor necrosis factor.

**Table 1 biomedicines-11-01097-t001:** NAFLD diagnostic scale.

Scale	Assessed Parameters	Diagnostic Value
NAFLD liver fat score (N-LFS) [68,69]	MS, T2DM, FSI, AST, and the AST/ALT ratio.	NLFS ≥ 0.640 (86% sensitivity and 71% specificity in identification of hepatic steatosis > 5.56%).
Fatty liver index (FLI) [70,71]	WC, BMI, TG, and GGTP.	FLI < 30 the lack of fatty liver (sensitivity 87%, specificity 64%), FLI ≥ 60 the presence of fatty liver (61% sensitivity and 86% specificity).
Hepatic steatosis index (HSI) [72,73]	Gender, history of T2DM, BMI, ALT, and AST.	HSI < 30 excludes NAFLD (92.5% sensitivity with 0.186 negative likehood ratio) and HSI > 36 detects NAFLD (92.4% specificity and 6.069 positive likehood ratio).
Lipid accumulation product (LAP) [74,75]	WC, TG, and gender.	The LAP values in men 30.5 (77% sensitivity, 75% specificity) and in women (%) 23.0 (82% sensitivity 82%, 79% specificity).
SteatoTest [76,77]	Serum α2-macroglobulin, apo A1, haptoglobin, total bilirubin, GGTP, ALT, BMI, TCH, TG, and glucose adjusted for age and gender.	0.30 with 90% sensibility and 0.72 with 90% specificity to diagnose hepatic steatosis in 2–4 grade.
NAFL screening score [78,79]	Age, FPG, BMI, TG, ALT/AST, and uric acid.	33 for men (80% sensitivity, 66% specifivity) and 29 for women (89% sensitivity, 69% specificity)

ALT—alanine aminotransferase, AST—aspartate aminotransferase, BMI—body mass index, FPG—fasting plasma glucose, FSI—fasting serum insulin, GGTP—gamma glutamyl transpeptidase, MS—metabolic syndrome, NAFL—non-alcoholic fatty liver, NAFLD—non-alcoholic fatty liver disease, T2DM—type 2 diabetes mellitus, TCH—total cholesterol, TG—triglycerides, WC—waist circumference.

## Data Availability

Not applicable.

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
