# Peer review of "Non-Alcoholic Fatty Liver Disease or Type 2 Diabetes Mellitus—The Chicken or the Egg Dilemma"

_biomedicines, 2023, doi:10.3390/biomedicines11041097_

Round 1
Reviewer 1 Report
Authors should put a clear distinction between the onset of T2DM and its diagnosis, because symptoms or clinical manifestations that lead physicians to diagnose T2DM through laboratory analyses could appear in a relatively advanced time. Thus, T2DM was only discovered in a successive occasion respect to NAFLD…and T2DM was not a consequence of NAFLD, but only a delayed/successive diagnosis.
All of this turns around the debate which one is the best definition of the chronic liver disease, i.e., NAFLD or MAFLD, as evident in.....NAFLD or MAFLD: That is the conundrum. Hepatobiliary Pancreat Dis Int. 2022 Apr;21(2):103-105. doi: 10.1016/j.hbpd.2022.01.008. Epub 2022 Jan 31. PMID: 35125337.
As correctly stated by authors there is still much criticism around the new proposed definition, i.e., MAFLD, due to the fact that not aways the two entities overlap.
Author Response
Response to Reviewer 1 Comments
Thank the Reviewer’s very much for your time and valuable comments on our manuscript. The responses for all points are below. The changes were introduced into the text of manuscript, as suggested by the Reviewer.
Point 1: Authors should put a clear distinction between the onset of T2DM and its diagnosis, because symptoms or clinical manifestations that lead physicians to diagnose T2DM through laboratory analyses could appear in a relatively advanced time. Thus, T2DM was only discovered in a successive occasion respect to NAFLD and T2DM was not a consequence of NAFLD, but only a delayed/successive diagnosis.
Response 1: Thank you for valuable remark. We added the following sentences in to epidemiological data (section 2, line 221-228) “T2DM is diagnosed at an advanced stage when patients report signs such as polyuria, polydipsia and weight loss. The onset of T2DM is known to be asymptomatic, although the carbohydrate metabolism disturbances are represented, however they are not recognized by the patients. Research show that this misdiagnosis affects a very high percentage of patients and is caused primarily by the asymptomatic onset of carbohydrate metabolism disorders, as well as by other factors such as the choice of a diagnostic tool, hypertension, non-cash insurance and >10 years of physician’s experience (42, 43)”
Point 2: All of this turns around the debate which one is the best definition of the chronic liver disease, i.e., NAFLD or MAFLD, as evident in.....NAFLD or MAFLD: That is the conundrum. Hepatobiliary Pancreat Dis Int. 2022 Apr;21(2):103-105. doi: 10.1016/j.hbpd.2022.01.008. Epub 2022 Jan 31. PMID: 35125337
Response 2: Thank the Reviewer’s suggestion. According to suggestion we used Tarantino, T. NAFLD or MAFLD: That is the conundrum. Hepatobiliary Pancreat Dis Int 2022, 2, 103-105.
In addition, we modified our article according to the suggestions of other reviewers.
We sincerely hope that all changes introduced by us in the text will be fully satisfactory for the Reviewer.
Reviewer 2 Report
The manuscript entitled "Non-alcoholic fatty liver disease or type 2 diabetes mellitus the chicken or the egg dilemma” led by Marcin Kosmalski et al., reviewed on an interesting dilemma topic to bring attention to researcher that coexistence of Non-alcoholic fatty liver disease (NAFLD) and type 2 diabetes mellitus (T2DM) are common. Authors review is the extension of their study that majority (70%) of their NAFLD patients had T2DM. Authors concluded the review with outstanding points. Although the review is current and the following points are written to improve the manuscript. \
Comments to improve:
Authors should consider adding sub heading to the introduction paragraphs and in all other sections
Page 3: authors should consider using morbidly obese instead of ‘giant’ obesity and also including BMI is the current clinical practice. For children, authors could use percentile ranking of weight and height to compare obesity
Page 3; 2nd paragraph:
a. It should be ‘liver disease-related end stage complications’.
b. Variceal banding is procedure to block variceal bleeding should be clarified as well
c. Need to add a citation for line 130.
d. Since the topic of the review is NAFLD, authors should consider removing the details of alcoholic liver disease (ALD) and including the details of MAFLD as ALD related.
Page 4: Authors could consider adding some the new data from CDC that different ethnic groups have variable incidence of obesity, which could be related to NAFLD and T2DM.
Line 245 and Line 353 should be NAFLD
Clinical view: Authors can consider adding their personal experience from their publication
Authors should consider citing original research articles, e.g. ref 86 is another review article published elsewhere.
Information on ER stress and related pro-inflammatory cytokine activation during NAFLD and T2DM can be further expanded in page 11 to increase readership.
The description on IRS phosphorylation is nicely done in line 525 unlike 513.
The review is heavily focused on NAFLD and including the mechanistic insights on NAFLD compared to T2DM. Since this, two conditions are coexisting and cannot be investigated in adverse conditions, but can be studied in milder disease condition. Authors could also include the details of recent clinical trials with obetecholic acid and others where the outcome changed with and without T2DM
Impaired insulin secretion due to beta cell lipoapoptosis theory has been challenged with very little evidence from the literature. This controversy should be discussed in detail to allow future perspectives on this area to graduate student’s readership
Generally, liver is considered as insulin sensitive organ even during T2DM, The mechanism hepatic insulin extraction requires further clarification
Finally, authors should consider including the following in their title “Mechanistic pathways in the development of NAFLD and T2DM”
Author Response
Response to Reviewer 2 Comments
Thank the Reviewer’s very much for your time and valuable comments on our manuscript. The responses for all points are below. The changes were introduced into the text of manuscript, as suggested by the Reviewer.
Point 1: Page 3: authors should consider using morbidly obese instead of ‘giant’ obesity and also including BMI is the current clinical practice. For children, authors could use percentile ranking of weight and height to compare obesity
Response 1: According to your suggestion, we used BMI to indicate the type of obesity, replace “giant obesity” instead of morbidly obesity, For children we used percentile ranking.
Point 2: Page 3; 2nd paragraph:
a. It should be ‘liver disease-related end stage complications’.
Response 2a: Thanks for attention, we corrected this phrase in the manuscript..
b. Variceal banding is procedure to block variceal bleeding should be clarified as well
Response 2b: Thanks for attention, the indicated sentence was clarified.
c. Need to add a citation for line 130.
Response 2c: The following reference was added “Golabi, P.; Paik, J.M.; Eberly, K.; de Avila, L.; Alqahtani , S.A.; Younossi, Z.M. Causes of death in patients with Non-alcoholic Fatty Liver Disease (NAFLD), alcoholic liver disease and chronic viral Hepatitis B and C. Ann Hepatol 2022, 27, 100556.”
d. Since the topic of the review is NAFLD, authors should consider removing the details of alcoholic liver disease (ALD) and including the details of MAFLD as ALD related
Response 2d: Thanks for prompt remark. We made the following changes in the manuscript (line 65-73) “Controversies related to the new definition of NAFLD/MAFLD result from consideration of all causes leading to excessive fatty infiltration of the liver (for example susceptibility gene polymorphism changes, obstructive sleep apnea syndrome, polycystic ovary syndrome). In addition, the discussion also arises from the need to take into account alcoholic fatty liver disease (ALD) as a protective factor for NAFLD and the need to maintain alcohol abstinence in NAFLD patients. For this reason, it is suggested that more than 10% of patients with MAFLD are not diagnosed as having NAFLD. It should therefore be emphasized that there are still many unknowns in the discussion related to the definition of NAFLD/MAFLD and it requires further research (10).”
Point 3: Authors could consider adding some the new data from CDC that different ethnic groups have variable incidence of obesity, which could be related to NAFLD and T2DM.
Response 3: Thank you for sugestion. We made introduced the following sentences in the manuscript (line 187-193) “The ethnicity also affects the frequency of NAFLD. According to the report by Huang et al. the Hispanic population had a higher prevalence of NAFLD (37.0%). However the non-Hispanic Black population had a lower prevalence of NAFLD (24.7%) compared to the non-Hispanic White subjects (29.3%) (28). Contrary recent data from the Centers for Disease Control and Prevention showed an increased incidence of T2DM in American Indians and Alaska Natives (14.5%), non-Hispanic Blacks (12.1%) and people of Hispanic origin (11.8%) (29).”
Point 4: Line 245 and Line 353 should be NAFLD
Response 4: Thank for attention. We corrected these oversights.
Point 5: Authors can consider adding their personal experience from their publication
Response 5: Thanks for suggestion. We described the results of our study in the section Epidemiological data - reference no 37.
Point 6: Authors should consider citing original research articles, e.g. ref 86 is another review article published elsewhere.
Response 6: Thank the Reviewer’s suggestion. According to it, we made the changes in the manuscript in place, where we can do it, for example reference Zhang, C.; Hwarng, G.; Cooper, D.E.; Grevengoed, T.J.; Eaton, J.M.; Natarajan, V.; Harris, T.E.; Coleman, R.A. Inhibited insulin signaling in mouse hepatocytes is associated with increased phosphatidic acid but not diacylglycerol. J Biol Chem 2015, 290, 3519-28.
Point 7: Information on ER stress and related pro-inflammatory cytokine activation during NAFLD and T2DM can be further expanded in page 11 to increase readership.
Response 7: Thank you for prompt remark. We replaced review papers by original papers in each possible place (line 568-591) “Furthermore, the accumulation of excessive amount of lipids in the liver causes lipotoxicity which interacts with ER stress and culminates in inflammation and hepatocellular damage, the mechanisms crucially implicated in NASH pathogenesis. Finally, the circadian clock machinery regulates ER stress-related pathways and vice versa, thus controlling the liver metabolism homeostasis and being implicated in the NAFLD progression (100). The accumulation of excess lipids in hepatocytes increases the demand for protein processing by the ER, causing misfolded proteins to accumulate in the ER lumen. ER stress is induced by excess misfolded or unfolded proteins, and the unfolded protein response (UPR) is triggered to restore homeostasis. UPR associated with membrane biosynthesis, insulin action, inflammation and apoptosis, serves to restore ER homeostasis by reduction protein synthesis and enhancement of protein folding and clearance. ER stress manifestation id prominent in inflammatory responses, including direct defense against microbial pathogens, production of pro-inflammatory cytokines, immunogenic cell death, metabolic homeostasis and maintenance of immune tolerance. In the course of these processes, immune cells infiltrate the liver and release pro-inflammatory cytokines and immunomodulatory mediators that may worsen the dysfunction of hepatocytes, resulting in hepatocyte necrosis, hepatic steatosis and fibrosis, which may result in NAFLD and NASH. On the other hand, the conditions most conducive to the progression of ER stress mediated disease may include chronic injury that induces persistent ER stress, which is associated with a reduced or impaired ability of the general immune response to mitigate inflammatory damage. At the onset of NASH, the damaged hepatocytes release a variety of signals, such as, among others, damage-associated molecular patterns and pathogen associated molecular patterns, which activate local and mobilized immune cells and trigger an immune response (101).”
Point 8: The description on IRS phosphorylation is nicely done in line 525 unlike 513.
Response 8: Thanks for remark. We removed text in line 591-594.
Point 9: The review is heavily focused on NAFLD and including the mechanistic insights on NAFLD compared to T2DM. Since this, two conditions are coexisting and cannot be investigated in adverse conditions, but can be studied in milder disease condition. Authors could also include the details of recent clinical trials with obetecholic acid and others where the outcome changed with and without T2DM.
Response 9: Thank the Reviewer’s suggestion. Raising this important topic of the use of obetecholic acid in patients with NAFLD would certainly enrich the work.
Point 10: Impaired insulin secretion due to beta cell lipoapoptosis theory has been challenged with very little evidence from the literature. This controversy should be discussed in detail to allow future perspectives on this area to graduate student’s readership.
Response 10: Thank for for valuable remark. We added the following stentences (line 676-684) “Furthermore, the accumulation of excessive amount of lipids in the liver causes lipotoxicity which interacts with ER stress and culminates in inflammation and hepatocellular damage, the mechanisms crucially implicated in NASH pathogenesis. Finally, the circadian clock machinery regulates ER stress-related pathways and vice versa, thus controlling the liver metabolism homeostasis and being implicated in the NAFLD progression (100). The accumulation of excess lipids in hepatocytes increases the demand for protein processing by the ER, causing misfolded proteins to accumulate in the ER lumen. ER stress is induced by excess misfolded or unfolded proteins, and the unfolded protein response (UPR) is triggered to restore homeostasis. UPR associated with membrane biosynthesis, insulin action, inflammation and apoptosis, serves to restore ER homeostasis by reduction protein synthesis and enhancement of protein folding and clearance. ER stress manifestation id prominent in inflammatory responses, including direct defense against microbial pathogens, production of pro-inflammatory cytokines, immunogenic cell death, metabolic homeostasis and maintenance of immune tolerance. In the course of these processes, immune cells infiltrate the liver and release pro-inflammatory cytokines and immunomodulatory mediators that may worsen the dysfunction of hepatocytes, resulting in hepatocyte necrosis, hepatic steatosis and fibrosis, which may result in NAFLD and NASH. On the other hand, the conditions most conducive to the progression of ER stress mediated disease may include chronic injury that induces persistent ER stress, which is associated with a reduced or impaired ability of the general immune response to mitigate inflammatory damage. At the onset of NASH, the damaged hepatocytes release a variety of signals, such as, among others, damage-associated molecular patterns and pathogen associated molecular patterns, which activate local and mobilized immune cells and trigger an immune response (101).”
Point 11: Generally, liver is considered as insulin sensitive organ even during T2DM. The mechanism hepatic insulin extraction requires further clarification.
Response 11: According to your comment we clarified mechanism of hapitc insulin extraction as followed (line 502-510) ” From pancreatic β-cells, insulin reaches the liver through the portal circulation to exert its action and eventually undergo clearance in the hepatocytes. The hepatic insulin clearance regulates the homeostatic level of insulin that is required to reach the peripheral insulin target tissues to elicit proper insulin action. Receptor-mediated insulin uptake followed by its degradation constitutes the basic mechanism of insulin clearance. Upon its phosphorylation by the insulin receptor tyrosine kinase, carcinoembryonic antigen-related cell adhesion molecule 1 (CEACAM1) becomes a part of the insulin-insulin receptor complex and increase the rate of its endocytosis and targeting to the degradation pathways (89).”
Point 12: Finally, authors should consider including the following in their title “Mechanistic pathways in the development of NAFLD and T2DM”
Response 12: Thank for you comment. We would like to keep our title if possible. In our opinion it reflects the topic of the paper.
In addition, we modified our article according to the suggestions of other reviewers.
We sincerely hope that all changes introduced by us in the text will be fully satisfactory for the Reviewer.